# Mammalian Glycosylation Patterns Protect Citrullinated Chemokine MCP-1/CCL2 from Partial Degradation

**DOI:** 10.3390/ijms24031862

**Published:** 2023-01-18

**Authors:** Olexandr Korchynskyi, Ken Yoshida, Nataliia Korchynska, Justyna Czarnik-Kwaśniak, Paul P. Tak, Ger J. M. Pruijn, Takeo Isozaki, Jeffrey H. Ruth, Phillip L. Campbell, M. Asif Amin, Alisa E. Koch

**Affiliations:** 1Department of Human Immunology and Centre for Innovative Biomedical Research, Medical Faculty, University of Rzeszow, 1a Warzywna St., 35-310 Rzeszów, Poland; 2Department of Clinical Immunology and Rheumatology, Academic Medical Center/University of Amsterdam, 1105 AZ Amsterdam, The Netherlands; 3Department of Molecular Immunology, Palladin Institute of Biochemistry, National Academy of Sciences of Ukraine, 01054 Kyiv, Ukraine; 4Department of Public Development and Health, S. Gzhytskyi National University of Veterinary Medicine and Biotechnologies, 79010 Lviv, Ukraine; 5Division of Rheumatology, University of Michigan Medical School, Ann Arbor, MI 48109, USA; 6Division of Rheumatology, Department of Internal Medicine, the Jikei University School of Medicine, Tokyo 105-8461, Japan; 7Department of Internal Medicine, University of Cambridge, Cambridge CB2 1TN, UK; 8Candel Therapeutics, Needham, MA 02494, USA; 9Department of Biomolecular Chemistry, Institute for Molecules and Materials, Radboud University, 6525 AJ Nijmegen, The Netherlands

**Keywords:** monocyte chemoattractant protein-1, post-translational modifications, glycosylation, citrullination, peptidylarginine deiminase, partial degradation

## Abstract

Monocyte chemoattractant protein-1 (MCP-1/CCL2) is a potent chemotactic agent for monocytes, primarily produced by macrophages and endothelial cells. Significantly elevated levels of MCP-1/CCL2 were found in synovial fluids of patients with rheumatoid arthritis (RA), compared to osteoarthritis or other arthritis patients. Several studies suggested an important role for MCP-1 in the massive inflammation at the damaged joint, in part due to its chemotactic and angiogenic effects. It is a known fact that the post-translational modifications (PTMs) of proteins have a significant impact on their properties. In mammals, arginine residues within proteins can be converted into citrulline by peptidylarginine deiminase (PAD) enzymes. Anti-citrullinated protein antibodies (ACPA), recognizing these PTMs, have become a hallmark for rheumatoid arthritis (RA) and other autoimmune diseases and are important in diagnostics and prognosis. In previous studies, we found that citrullination converts the neutrophil attracting chemokine neutrophil-activating peptide 78 (ENA-78) into a potent macrophage chemoattractant. Here we report that both commercially available and recombinant bacterially produced MCP-1/CCL2 are rapidly (partially) degraded upon in vitro citrullination. However, properly glycosylated MCP-1/CCL2 produced by mammalian cells is protected against degradation during efficient citrullination. Site-directed mutagenesis of the potential glycosylation site at the asparagine-14 residue within human MCP-1 revealed lower expression levels in mammalian expression systems. The glycosylation-mediated recombinant chemokine stabilization allows the production of citrullinated MCP-1/CCL2, which can be effectively used to calibrate crucial assays, such as modified ELISAs.

## 1. Introduction

Chemokines are members of the chemoattractant cytokines with molecular weights from 8 to 12 kDa, divided into four subfamilies (C, CC, CXC and CX3C), based on the position and spacing of their N-terminal cysteine residues [1,2,3]. These cytokines have a unique ability to recruit and activate monocytes, neutrophils and lymphocytes [4]. The monocyte chemoattractant proteins (MCPs) constitute an important group within the CC-chemokine sub-family. Five MCPs have been named and classified: MCP-1/CCL2, MCP-2/CCL8, MCP-3/CCL7, MCP-4/CCL13 and MCP-5/CCL12.

In humans, MCP-1/CCL2 was the first described and is the best characterized to date. The *CCL2* gene is located on chromosome 17 and encodes a 13 kDa protein composed of 76 amino acids [4,5]. MCP-1/CCL2 is produced by different cell types, including macrophages, dendritic cells and endothelial cells. MCP-1/CCL2 is well known for its activities in the immune context: it exerts chemotactic activity for several cell types, such as monocytes, T lymphocytes, NK cells, basophils and induces directed leukocyte motility. The chemokine stimulates the migration of T cells to infected or damaged sites [6,7]. MCP-1/CCL2 was found in the synovium of patients with rheumatoid arthritis (RA), gout, traumatic arthritis, atherosclerosis, multiple sclerosis and various cancers [8,9]. Several studies have shown increased levels of MCP-1/CCL2 in synovial tissue and fluids, as well as in the sera and plasma of RA patients versus osteoarthritis or other arthritis patients [10,11].

The post-translational citrullination modification of proteins is based on the enzymatic conversion (deimination) of arginine (Arg) to citrulline (Cit). Such conversion leads to very minor changes in the molecular mass of the MCP-1/CCL2 protein, but causes loss of a positive charge [12,13]. This post-translational conversion into citrulline is catalysed by peptidylarginine deiminases (PADs). In mammals, five isotypes have been described: PAD1-4 and PAD6. PAD enzymes are generally found in the cytoplasm of various cell types, except for PAD4 which can translocate to the nucleus [14,15]. Protein citrullination occurs during the physiological ‘maturation’ of the skin, while its aberrant increase is linked to various diseases, including cancer, lupus, multiple sclerosis, and, as mentioned, RA. This last disorder constitutes an autoimmune disorder with a high morbidity, often leading to the progressive destruction of the joints, resulting in pain and stiffness. Rheumatic joints are rich with pro-inflammatory cytokines, including tumor necrosis factor-α (TNF-α), interleukin 1 and 6, granulocyte-macrophage colony-stimulating factor and chemokines, such as interleukin-8 [16]. Nowadays, anti-citrullinated peptide antibodies (ACPAs) are considered a hallmark of the disease and routinely used to diagnose it [17].

Glycosylation represents another well-known, highly heterogeneous, post-translational modification of proteins, involved in folding, protease protection and protein-protein contact. N-linked glycosylation refers to the co-translational attachment of sugars to the nitrogen atom of asparagine residues, concentrated on the exterior of mature proteins. During maturation, it also contributes to protein folding. O-glycosylation of proteins is a post-translational event that takes place subsequently to N-glycosylation and protein folding. It refers to the O-linked attachment of sugars to serine and threonine, and, to a lesser extent, hydroxyproline and hydroxylysine. Human and mouse CC chemokines may be glycosylated. MCP-1/CCL2 is expressed as an 8 kDa protein which contains a short N-terminal secretion signal. During processing, this signal peptide is removed, thus allowing the N-terminal glutamate residue to be cyclised to pyroglutamate. MCP-1/CCL2 also contains two consensus glycosylation sequences, one for N-glycosylation in the N-terminal region (asparagine residue at position 14 in mature MCP-1/CCL2 protein and another for O-glycosylation). Ruggiero et al. suggested the possibility that the majority of human and mouse MCP-1/CCL2 remains unprotected against proteases because of inefficient glycosylation [3,18,19].

Recombinant human MCP-1/CCL2 has been expressed in many different expression systems, such as *Escherichia coli*, the methylotrophic yeast *Komagataella* (previously referred to as *Pichia) pastoris,* COS-1 fibroblast-like cells (derived from monkey kidney tissue), or insect cells [3]. Depending on the type of the expression system used, differences in protein activity were observed. For example, if *E. coli* is used for expression, the recombinant human MCP-1/CCL2 is unstable, but active, and similar observations were made after expression in insect cells or *K.pastoris* [3]. In all the cases, protein stability remains an issue.

The main goal of our study is to establish an efficient procedure for the preparation of stable citrullinated chemokine MCP-1/CCL2, which would be highly useful for different research applications. In light of a fact that we detected a quick loss of bacterially produced MCP-1 upon in vitro citrullination, we surmised that citrullinated chemokines can be stabilized and protected from quick degradation by other post-translational modifications in mammalian cells.

## 2. Results

The main goal of our work was to generate a standard procedure to prepare stable citrullinated MCP-1/CCL2 chemokine suitable for research applications. We amplified the sequences encoding full length and mature versions of MCP-1/CCL2 from cDNA prepared from IL-1β-treated primary synovial fibroblasts derived from RA patients. A mature version equivalent to completely processed MCP-1/CCL2, was cloned into a frame with a C-terminal 6xHis-tag into pET19b bacterial expression vector that allows expression of the insert in bacterial cells. 

Commercially available recombinant MCP-1/CCL2 from two different suppliers (Peprotech, United Kingdom and R&D Systems, Minneapolis, MN, USA) as well as self-made bacterially produced MCP-1/CCL2 were citrullinated in vitro using recombinant human PAD2, PAD4 or rabbit PAD2. Reaction products were analyzed by SDS-PAGE with subsequent colloidal Coomassie staining (Figure 1a) or western blotting (Figure 1b).

Overall, we failed to detect robust citrullination of MCP-1/CCL2. First, we found that both self-made bacterially produced MCP-1/CCL2 and commercially available bacterially-produced recombinant MCP-1/CCL2 from two different producers (Peprotech, United Kingdom and R&D Systems, Minneapolis, MN, USA.) were quickly degraded during the in vitro citrullination; see a representative example in Figure 1b. 

Therefore, the 6xHis-tagged version of MCP-1/CCL2 chemokine cDNA was further re-cloned into mammalian pcDEF expression vector, in an order to confer a high expression level of MCP-1/CCL2 chemokine. The mammalian expression vector was transfected into HEK 293T cells using a PEI reagent. The cellular lysate with glycosylated mature MCP-1/CCL2 was collected and purified with ProBond nickel beads and gradually eluted with 50, 100, 150 and 200 mM imidazole. Both the pre-protein and its mature version contain four arginine residues at the positions corresponding to 18, 24, 29 and 30 in the mature protein (Figure 2a). The diagnostic 1-Da mass shift occurring upon citrullination of MCP-1/CCL2, was identified by mass spectrometry at the peptide that contains Arg-18 of the mature protein (Figure 2b). In spite of the presence of three more arginine residues at the positions 24, 29 and 30 within the mature MCP-1 protein (Figure 2a), we failed to detect citrullination at those residues using a mass-spectrometry. 

The quality and quantity of in vitro citrullinated *E. coli* expressed recombinant protein purchased from R&D systems (Figure 3a) was compared with the self-made recombinant protein produced in HEK293T cells (Figure 3b) using an ELISA specific for this protein. Interestingly, upon successful citrullination (Figure 1 and Figure 2) all of the preparations (two commercially produced ones and multiple aliquots of self-made prep) of bacterially produced chemokines appeared to be very unstable: we were able to detect some random remains of recombinant proteins in such preps, exclusively immediately after the in vitro citrullination reaction. All of the citrullinated bacterial proteins were undetectable upon storage under −70 °C. A suitability of the mammalian cells-produced chemokine standards was confirmed with a high coefficient of determination (R^2^ = 0.99386, Figure 3b) versus the low R squared value for the bacterially produced chemokine (R^2^ = 0.70629, Figure 3a). Thus, we have shown that in vitro citrullinated mammalian cell-produced (Figure 3b), but not the bacterially produced MCP-1/CCL2 chemokine (Figure 3a) can be efficiently used as the good quality standards in ELISA.

In silico analysis of MCP-1 reveals a potential glycosylation site at the asparagine residue at position 14 (N-14, Figure 4a). To confirm the importance of glycosylation at this mentioned residue, we performed a site-directed mutagenesis aiming to convert asparagine into a chemically very similar glutamine residue, which still cannot serve as a docking site for glycosylation. A western blot analysis revealed both wild type and mutant recombinant proteins hMCP-1/CCL2 (Figure 4b). At the same time, the expression level of N14Q mutant hMCP-1/CCL2 protein was significantly lower than of wild-type protein. As expected, the glycosylation slightly shifted a recombinant wild-type protein (for around 2 kDa), in a comparison with bacterially produced chemokine (Figure 4b). To compare the stability of wild-type and N14Q mutant recombinant hMCP-1/CCL2 upon their citrullination, we performed a citrullination of different MCP-1 protein versions with recombinant human PAD2/3/4 enzymes overexpressed and produced into HEK293T cell (Figure 4b). Similarly to the described above results (Figure 1), a hPAD2/3/4 led to the destruction of bacterially produced hMCP-1/CCL2. In the case of PAD2 and PAD4, such destruction was practically complete, while for PAD3 it was partial (Figure 4b). The mammalian cell-produced recombinant hMCP-1/CCL2 was successfully detected upon treatment with all three PAD enzymes, albeit at slightly lower levels, than the protein untreated with PAD (Figure 4b).

## 3. Discussion

It is well known that post-translational modifications affect the chemical and biological properties of proteins. For example, glycosylation modulates the specific biological and biochemical activities of some cytokines [22], including chemokines. MCP-1/CCL2, MCP-3/CCL7 and HCC-1 have been demonstrated to be glycosylated in vivo [19]. MCP-1/CCL2 is one of the first identified chemokines, which can be detected in both glycosylated and non-glycosylated forms [23]. Rutledge et al., 1995 have shown that glycosylated MCP-1/CCL2 was two- to threefold less chemotactic for monocytes and THP-1 cells than non-glycosylated MCP-1/CCL2 [22]. In the case of CCL5 and CCL11 chemokines, there were no differences in in vitro eosinophil chemotactic activity between glycosylated and non-glycosylated forms [24]. Dong et al. used insect cells to produce recombinant XCL1 chemokine. In their chemotaxis assays performed with primary CD4^+^ T cells, recombinant, glycosylated, the XCL1 chemokine was more potent when compared to the non-glycosylated form and it was a more effective inhibitor of T cell proliferation [19]. In our current study, we are demonstrating that the glycosylation process strongly stabilizes citrullinated MCP-1/CCL2 and protects it against gradual degradation, which occurs with a bacterially (*E. coli)* produced purified recombinant protein. It can be inferred that glycosylation of the CC-class chemokine MCP-1/CCL2, as well as of other chemokines (in our studies validated for CCL3/MIP-1α) protects the molecules against proteases. In turn, recombinant mouse CCL2 expressed in CHO cells, is highly glycosylated at the C-terminus with mainly O-linked sugars and this glycosylation is not essential for chemotaxis induction [18]. Through O-glycosylation, the different molecular masses of MCP-1/CCL2 were shown, which additionally contributed to the chemotactic potency reduction [25]. 

Currently, there are different systems for the expression of recombinant proteins. By using the yeast *Komagataella* (earlier referred to as *Pichia*) *pastoris* expression system Ruggiero et al., 2003 have shown that human recombinant MCP-1/CCL2 glycosylated in yeast was 4–20 times less active in a chemotactic assay in vitro, compared to non-glycosylated *E. coli* expression system [3]. In the current study, we have also presented a tool for MCP-1/CCL2 chemokine expression in mammalian HEK 293T cells. In our model, we found that glycosylation dramatically stabilizes the chemokine and only mammalian properly glycosylated CCL2/MCP-1 can be efficiently used for in vitro citrullination reactions. Our data are still insufficient to draw clear conclusions, if such instability of bacterially produced chemokines upon their citrullination in vitro, is an intrinsic feature of this and other chemokines. Even highly purified commercial chemokines or PAD2 enzyme preparations potentially still can have some undetectable contamination with proteinases. Most likely, only artificial synthesis of the citrullinated versus non-citrullinated protein could provide such an answer.

PAD enzymes specificity were the subject of previous studies. We compared the sequence of human MCP-1/CCL2 with proposed optimal sites for human PAD2 and PAD4 enzymes [21]. However, neither neighboring amino acid residues of successfully citrullinated arginine at position 18 (Figure 2), nor for arginine residues at positions 24, 29 and 30 shared any significant similarity to the predicted optimal recognition site for both human PAD2 or PAD4 [21].

Most of the detailed studies devoted to MCP-1/CCL2 glycosylation were performed with the murine protein [26, 27}. Their results show a much more heavily glycosylated protein of 25–30 kDa [26,27], than 12–14 kDa 6His-tagged protein in our studies (Figure 1 and Figure 4). Such results suggest multiple sites of additional O-linked glycosylation in murine MCP-1/CCL2, than we saw for human protein (Figure 4b). However, studies performed with recombinant human MCP-1/CCL2 [28] revealed a protein product of size comparable with a protein in our studies (Figure 4b). In addition, the study of Needham et al., 1996 [28} suggests the presence of both glycosylated and nonglycosylated forms of recombinant human protein upon its production in mammalian expression systems. Such potential partial presence of a nonglycosylated protein form in our recombinant human MCP-1/CCL2 can explain why we also lost some portion of the recombinant HEK293 cell produced protein upon its citrullination by PAD enzymes in vitro (Figure 4b). At the same time, our data suggest the crucial importance of N-glycosylation at asparagine-14 for the stability of recombinant citrullinated MCP-1 (Figure 4b).

## 4. Materials and Methods

### 4.1. Reagents

Rabbit PAD2 was purchased from Sigma-Aldrich/Merck (Utrecht, The Netherlands). Recombinant MCP-1/CCL2 duo-kit (ELISA) and monoclonal mouse antibody specific for MCP-1/CCL2 were purchased from R&D Systems (Minneapolis, MN, USA). pET19b vector was purchased from Novagen/Merck-Millipore (Utrecht, The Netherlands). pDEF vector was kindly provided by Dr. Goldman.

### 4.2. Cell Culture

The studies were performed using immortalized human embryonic kidney 293T cells (HEK 293T) purchased from the American Type Culture Collection (ATCC; Bethesda, MD, USA). The cells were cultured in Dulbecco’s Modified Eagle’s Medium (DMEM; Biowest, Nuaillé, France), supplemented with 10% (*v*/*v*) fetal bovine serum (FBS; Biowest, Nuaillé, France) and 1% (*v*/*v*) penicillin/streptomycin (Biowest, Nuaillé, France). Cells were routinely split twice a week, when ~80% confluency was reached. The culture was maintained in a humidified atmosphere of 95% air and 5% CO_2_ at 37 °C.

### 4.3. MCP-1/CCL2 cDNA Cloning

Full-length MCP-1/CCL2 complementary DNA (cDNA) was prepared from total mRNA isolated from primary synovial fibroblasts derived from a patient with RA. Tissue samples were obtained after approval by the AMC Institutional Review Board and provision of informed consent by the subjects. The cDNA fragments encoding mature protein were PCR-amplified with the following primers (forward and reverse, respectively): 5′-TAATCCATGGGA-CAGCCAGATGCAATCCAATGCC-3’ and 5´-TAAGAATTCTCAGTGATGGTGATG-GTGATGAGTCTTCGGAGTTTGGGTTTG-3´. We incorporated a *Nco*I restriction site in the forward primer and an endogeneous stop codon with six preceding codons encoding histidines and the *Eco*RI restriction site in the reverse primer. All sequences were verified using BigDye Terminator sequencing (Life Technology/Thermo Fisher Scientific, Waltham, MA, USA). The 77 amino acid long mature version of MCP-1/CCL2 was cloned in frame with C-terminal 6xHis-tag into a pET19b expression vector which allows to express the insert in bacteria. For optimization of expression in mammalian cells, the insert containing a C-terminus 6xHis-tag was later re-cloned into a pcDEF expression vector [1,29].

### 4.4. Transformation of E. coli BL21 and Purification of Recombinant Human MCP-1/CCL2

The BL21-Rosetta (Novagen\Merck, Berlington, MA, USA) strain of *E. coli* was used for the transformation with bacterial expression vector pET19b to produce MCP-1/CCL2 chemokine. A single colony of bacteria cells was grown in 2 L of LB medium at 37 °C until reaching 0.6 at OD_600_, 1mM IPTG was then added to induce the protein expression for 6 h, at 30 °C. Following the induction, cells were harvested by centrifugation and resuspended into the ice-cold buffer consisting of: 150 mL 50mM sodium phosphate (pH8.0), 300 mM NaCl, 10 mM imidazole, 1 mM β-mercapto-ethanol, 10 µM leupeptin, 10 µM pepstatin, 10 µM aprotinin and 10 µM phenylmethylsulfonyl fluoride (PMSF, Sigma-Aldrich\Merck, Berlington, MA, USA). The suspension was sonicated on wet ice with 10 × 1s-long pulses at maximal energy and target protein from the extracts was collected and purified using ProBond nickel beads (Life Technologies/Thermo Fisher Scientific, Waltham, MA, USA), as described below.

### 4.5. Transfection of HEK 293T Cells and Purification of Recombinant Human MCP-1/CCL2

HEK 293T were seeded at 50% confluency in 15 cm Petri dishes. Next, the cells were transfected with the pDEF vector expressing 6xHis-tagged MCP-1/CCL2. Transfection was carried out using transfection reagent polyethyleniminе (PEI; Polysciences Inc., Warrington, PA, USA) following the manufacturer’s protocol. Following transfection (5–6 h), culture medium was replaced with fresh complete medium. Then, 48 h after transfection, the cells were lysed with lysing buffer containing 1% Triton X-100, 150 mL 50mM sodium phosphate (pH8.0), 300 mM NaCl, 10 mM imidazole, 1mM β-mercaptoethanol, as well as 10 µM leupeptin, 10µM pepstatin, 10 µM aprotinin and 10 µM PMSF. The 6xHis-tagged MCP-1/CCL2 from HEK 293T cell lysates was purified upon immobilization on ProBond nickel beads (Life Technologies, Carlsbad, CA, USA), rinsed with 10 mM imidazole added to the same lysis buffer and eluted gradually with 50 mM, 100 mM, 150 mM and 200 mM imidazole. The quality and quantity of the expressed recombinant MCP-1/CCL2 was assessed with the DuoSet ELISA kit (R&D Systems, Minneapolis, MN, USA), specific for this chemokine and with colloidal Coomassie staining after the sodium dodecyl sulfate-polyacrylamide gel electrophoresis (SDS-PAGE) resolution [1]. 

### 4.6. In Vitro Citrullination of MCP-1/CCL2

Then, the concentration of purified MCP-1/CCL2 was measured, 100 microliters of purified recombinant human chemokine (~100 ng/mL) was incubated with 0.5 units of rabbit skeletal muscle PAD (Sigma-Aldrich\Merck, Berlington, MA, USA) in 40 mM Tris-HCl, pH 7.6, 10 mM calcium chloride and 2.5 mM dithiothreitol for 2 h at 37 °C. Deimination was stopped with 25 mM EDTA. The diagnostic 1-Da mass shift occurring upon citrullination was identified by liquid chromatography tandem mass spectrometry (MS Bioworks, Ann Arbor, MI, USA) [1].

### 4.7. Site-Directed Mutagenesis

The targeted change of a glycosylation site at asparagine 14 into a glutamine was performed with a Quik-change method (Stratagene, La Jolla, CA, USA) essentially as before [29], using forward CTGCTGTTAT-cAa-TTCACCAATAG and reverse CTATTGGTGAA-tTg-ATAACAGCAG primers. The success of mutagenesis was confirmed with a sequencing of resulting plasmids.

### 4.8. Western Blotting

Western blotting was performed as described previously [20]. A monoclonal mouse antibody specific for human MCP-1/CCL2 (BioLegend, San Diego, CA, USA) was used at a dilution of 1:1000. The procedure for citrulline modification was described before [1,20]. Rabbit polyclonal antibody against modified citrulline [20] was used at a dilution of 1:1000. Secondary horseradish peroxidase-conjugated goat anti-rabbit IgG antibody (Amersham-Pharmacia\Merck, Berlington, MA, USA) was used in a 10^4^-fold dilution. Detection was performed by enhanced chemoluminescence (ECL; Thermo Fisher Scientific, Waltham, MA, USA). To increase the quality of the western blot detection for 293T cell produced recombinant hMCP-1, an immunoprecipitation with anti- human MCP-1/CCL2 antibody was performed.

### 4.9. Detection of Recombinant Human MCP-1/CCL2 by a Modified Enzyme-Linked Immunosorbent Assay (ELISA)

The anti-modified citrulline ELISA-based method uses a commercial ELISA kit developed for the detection of total chemokines in a combination with Senshu’s antibody that recognizes the modified citrullines. Briefly, ninety-six-well plates (Thermo Fisher Scientific, Waltham, MA, USA) were coated overnight at room temperature (RT) with a mouse anti-human MCP-1/CCL2 antibodies (R&D Systems, Minneapolis, MN, USA). Following each step, the plates were washed with phosphate buffered saline (PBS) which contained 0.05% of Tween20. The plates were then blocked with 1% bovine serum albumin (BSA) in PBS for 1 h at RT and incubated with samples or standards for 2 h at RT. Citrullinated recombinant human MCP-1/CCL2 and native–noncitrullinated recombinant human MCP-1/CCL2 purified from the transformed *E. coli* BL21 and transfected HEK 293T cells were used as standards and negative controls, respectively, for the ELISA. The samples were crosslinked on the plate with 1% glutaraldehyde in PBS for 30 min at RT [1]. 

The plates were then incubated with 0.2 mM Tris-HCl (pH 7.8) for 30 min at RT to block the crosslinking. Subsequently, the plates were incubated overnight at 37 °C in a citrulline-modification solution consisting of 2 parts of solution A that contained 0.0025% (*w*/*v*) iron (III) chloride, 4.6 M sulfuric acid and 3.0 M phosphoric acid; 1 part of solution B that contained 1% diacetylmonoxime, 0.5% antipyrine and 1M acetic acid; and 1 part of water [1,21]. Next, the plates were incubated for 2 h at RT with rabbit anti-modified citrulline antibodies (Merck-Millipore/Sigma-Aldrich, USA), diluted 1:2500 in PBS containing 1% BSA. The plates were then incubated for 2 h at RT with horseradish peroxidase (HRP)-conjugated swine anti-rabbit IgG (Dako/Agilent, Santa Clara, CA, USA), diluted 1:1,000 in PBS containing 1% BSA. Biotin-Tyramide Reagent (PerkinElmer, Waltham, MA, USA), diluted 1:1000 in 0.05 M tris-base (pH 8.5), was added, followed by HRP-conjugated streptavidin [1]. The classic colour reaction detection was performed, according to the manufacturer´s protocol.

### 4.10. Statistical Analysis

All quantitative experiments were repeated at least three times in duplicate. Concentration determinations and data plotting were performed in Microsoft Excel. The coefficient of determination R^2^ was calculated on the free web calculator EasyCalculations.com website (https://www.easycalculation.com/statistics/r-squared.php (accessed on 7 December 2022 at 1.22 a.m)).

## 5. Conclusions

Glycosylation is lacking in bacterially-produced proteins but is present in mammalian cells stabilized citrullinated MCP-1/CCL2. This process protects the chemokine against rapid partial degradation. Moreover, properly glycosylated MCP-1/CCL2 produced by mammalian cells can be further citrullinated in vitro and efficiently used as a standard in citrullination-specific ELISAs, as well as in further biological investigations.

## Figures and Tables

**Figure 1 ijms-24-01862-f001:**
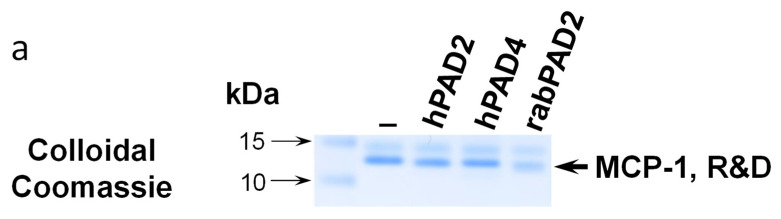
In vitro citrullination of bacterially produced chemokines leads to their partial degradation. (**a**) Colloidal Coomassie stained SDS-PAGE analysis of the MCP-1/CCL2 products of in vitro citrullination. In vitro citrullination of commercially available recombinant MCP-1/CCL2 and self-made bacterially produced MCP-1/CCL2 performed with recombinant human PAD2 and PAD4 or rabbit PAD2, respectively. Colloidal Coomassie stained MCP-1 bands are shown while recombinant humanPAD2/4 or rabbit PAD2 (Sigma Aldrich/Merck, Waltham, MA, USA) were added to reactions into the quantities below the Colloidal Coomassie sensitivity limits and cannot be visualized. (**b**) Immunoblot analysis of bacterially produced MCP-1/CCL2 upon an in vitro citrullination reaction. Self-made full-length bacterially produced MCP-1/CCL2 was citrullinated in vitro with rabbit PAD2 and resolved on SDS-PAGE, stained with Ponceau S. detection of total MCP-1/CCL2 and modified citrullines with Senshu’s antibody that recognizes that modified citrullines were made according to Senshu’s protocol [20]. Results shown are a representative of three or more repetitive experiments.

**Figure 2 ijms-24-01862-f002:**
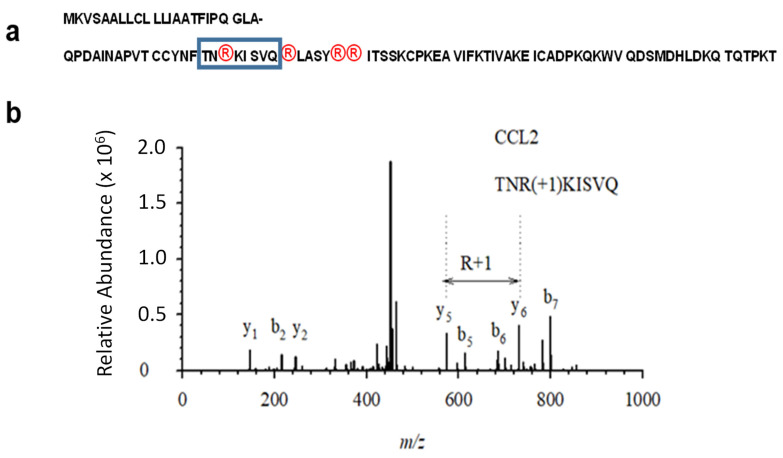
Mass-spectrometry verification of successful in vitro citrullination of MCP-1/CCL2. (**a**) Amino acid sequence of MCP-1/CCL2 (mature form of the protein spans residues 24–99). The first row shows 23 amino acid-long signaling peptide and the second row–mature protein of 76 residues. All arginine residues are highlighted with circles. The boxed areas show the peptide that contains the citrullinated arginine residues that were detected. (**b**) Annotated tandem mass spectrometry (MS/MS) fragmentation spectrum for the citrullinated MCP1/CCL2 peptide, showing the citrullinated arginine residue at position 3 of the studied peptide (R45). The MS/MS fragmentation data were annotated using Expert System (Max Planck Institute of Biochemistry). The precursor ion was observed with a mass error of 1 part per million, and the error for the fragment ions was 0.02 daltons.

**Figure 3 ijms-24-01862-f003:**
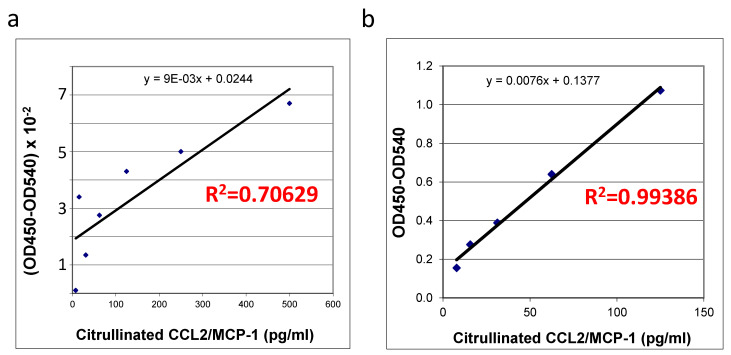
Detection of citrullinated recombinant human MCP-1/CCL2. Standard curves of citrullinated recombinant human MCP-1/CCL2 were set up using enzyme-linked immunosorbent assay in duplicate (**a**) standard curve of citrullinated *E. coli* produced recombinant human MCP-1/CCL2 chemokine (R&D Systems), (**b**) standard curve for citrullinated MCP-1/CCL2 chemokine produced by and purified from human HEK 293T cells. Absorbance at 450 nm is shown. Coefficients of determination R^2^ are indicated in red on the relevant plots. Results shown are a representative of three or more repetitive experiments.

**Figure 4 ijms-24-01862-f004:**
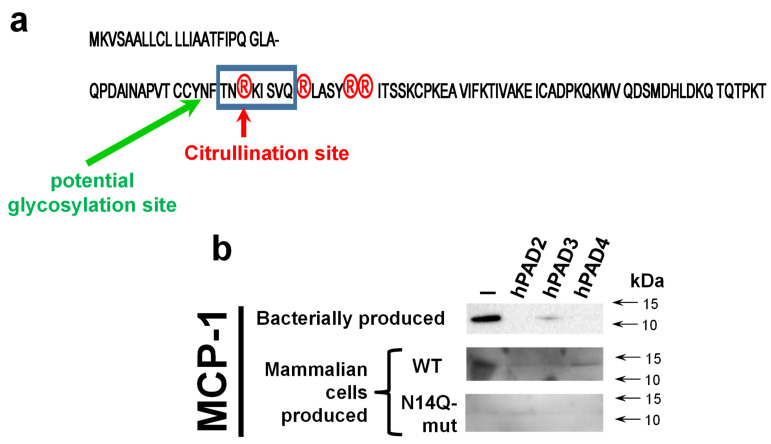
Site-directed mutagenesis of potential glycosylation site at asparagine-14 in mammalian cell-produced MCP-1 significantly destabilizes recombinant human MCP-1/CCL2. (**a**) Amino acid sequence of MCP-1/CCL2 with indicated positions of predicted N-glycosylation and citrullination site confirmed by a mass-spectrometry. (**b**) Immunoblot analysis of bacterially versus mammalian cell-produced MCP-1/CCL2 upon an in vitro citrullination reaction. Bacterially produced wild-type MCP-1/CCL2 or HEK293T cell-produced wild-type and N14Q mutant version of chemokine were incubated with EDTA-free proteinase inhibitor cocktail supplemented cellular lysates prepared from the control (vehicle-transfected) or indicated hPAD enzyme-transfected HEK293T cells. Citrullination reactions that were performed directly within cellular lysates essentially as published before [21]. Recombinant chemokine was concentrated with immunoprecipitation and resolved on 10–18% polyacrylamide gradient SDS-PAGE. Results shown are a representative of three experiments.

## Data Availability

Representative research data demonstrated into the manuscript, any data not shown can be provided upon special request.

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
