# Peer review of "Mammalian Glycosylation Patterns Protect Citrullinated Chemokine MCP-1/CCL2 from Partial Degradation"

_ijms, 2023, doi:10.3390/ijms24031862_

Round 1
Reviewer 1 Report (Previous Reviewer 1)
Authors adressed my concerns.
Reviewer 2 Report (Previous Reviewer 2)
In this revised manuscript, the authors have addressed all my questions and most of my concerns have been answered satisfactorily. I have no further comments on it.
This manuscript is a resubmission of an earlier submission. The following is a list of the peer review reports and author responses from that submission.
Round 1
Reviewer 1 Report
Korchynskyi et al. report that MCP-1/CCL2 produced by mammalian cells can be citrullinated in vitro without concomitant degradation, while this is not observed for commercially available or bacterially produced CCL2. This may have interesting application for the diagnosis of RA, as citrullinated CCL2 can be used to evaluate anti-citrullinated Abs, which are a diagnostic tool for this disease. However, the manuscript contains very few experimental data, and I would require including some additional experiments before considering it for publication:
-) Figure 1 only reports some of the data indicated in the manuscript. Authors stated to have tested three different proteins: R&D, Peprotech and bacterially produced (BP). Data in Figures 1a and 1b should include all three proteins and not just a representative case (also considering that few data are reported in the paper and therefore there is not any problem of exceeding manuscript or figure length).
-) Figure 2 should include western blot data to show mammalian cells-produced (McP) citrullinated CCL2 (as shown for figure 1).
-) Figure 3a,b should indicate the adjusted R2 value (and corresponding p-value) for each curve and not just the correlation coefficients. Otherwise, how we could identify a significant improvement in the quality of the obtained standard curve?
-) Data in Figure 3 indicate that McP citrullinated CCL2 could (but R2 and p-value are required to be sure) be more reliable than BP as a standard for anti-citrullinated Ab titration in ELISA. To strengthen this point, a functional test should be performed. For example, the chemoattractant activity of McP and BP citrullinated (or not) should be tested using T lymphocytes or monocytes (using several cell types would be better) using several doses of proteins (also performing a comparison between freshly prepared and after freeze-thawn cycles).
-) In the discussion, the authors state that the glycosylation of CCL2 in mammalian cells is important to stabilize the protein upon citrullination (something that is obviously not possible in bacteria). However, to state this point, authors should demonstrate that the glycosylation state of CCL2 is indeed important for its stability upon citrullination. For example, mutant CCL2 with mutated Asp 14 (which should not be glycosylated, to be verified and shown) could be tested against unmutated CCL2.
Minor Points:
-) Spelling error at line 33 (important in for diagnostic).
-) Spelling error at line 211 (equivalent to completely processed to completely processed MCP-1/CCL2).
Author Response
Dear Editor,
Dear Reviewers
We have greatly appreciated the very useful comments on our manuscript and performed additional experiments pointed out by the Reviewers. We believe, that we properly addressed all the Reviewers concerns and clearly explained all the unclear points. We hope, that the current version of our manuscript is greatly improved and now can be considered for publication in IJMS.
Specific concerns raised by Reviewers are pasted below with our replies point-by-point:
Reviewer 1 comments:
-) Figure 1 only reports some of the data indicated in the manuscript. Authors stated to have tested three different proteins: R&D, Peprotech and bacterially produced (BP). Data in Figures 1a and 1b should include all three proteins and not just a representative case (also considering that few data are reported in the paper and therefore there is not any problem of exceeding manuscript or figure length).
As one can see from our Fig.1, we included data obtained from the analysis of bacterially produced protein from R&D Systems and self-made preps. We optionally did not include data with Peprotech protein preps because they revealed a strong discrepancy between declared quantities of the recombinant protein and actual protein content. Below we attached the data of Colloidal Coomassie staining. We loaded 250 ng of recombinant protein per lane (500 ng totally per two lanes) of five Peprotech chemokines: CCL2/MCP1, CCL3/Mip1α, CXCL5/ENA78 CXCL8/IL-8 and CXCL12/SDF1. The detection sensitivity for Colloidal Coomassie is predicted to be at least 100 ng. We loaded 2,5 times more recombinant protein per lane and only in one case (CXCL8/IL-8) we found very weak staining. Neither of the Peprotech products were detected upon in vitro citrullination (pls., see below). Due to high price of the purified recombinant proteins we switched to a more reliable producer (R&D Systems) for which we easily and readily detected the declared amounts of the recombinant proteins while using the exact same prep of Colloidal Coomassie reagent and following exactly the same experimental protocol.
We prefer to avoid any discussion of the quality of Peprotech products in our manuscript as such a point is distant of our primary goals of the study and politely hope Reviewer 1 will understand.
-) Figure 2 should include western blot data to show mammalian cells-produced (McP) citrullinated CCL2 (as shown for figure 1).
We showed the mass-spectrometry confirmation of successful citrullination. Therefore, we believe that a Western blot confirmation would be redundant. We have shown successful citrullination of ENA-78 previously (Yoshida et al., A&R-2014). In a current study, we also used dot blot confirmation.
-) Figure 3a,b should indicate the adjusted R2 value (and corresponding p-value) for each curve and not just the correlation coefficients. Otherwise, how we could identify a significant improvement in the quality of the obtained standard curve?
According to Reviewer1’s comments we performed an additional statistical analyses. Calculated R2 values are indicated on each relevant plot.
-) Data in Figure 3 indicate that McP citrullinated CCL2 could (but R2 and p-value are required to be sure) be more reliable than BP as a standard for anti-citrullinated Ab titration in ELISA. To strengthen this point, a functional test should be performed. For example, the chemoattractant activity of McP and BP citrullinated (or not) should be tested using T lymphocytes or monocytes (using several cell types would be better) using several doses of proteins (also performing a comparison between freshly prepared and after freeze-thawn cycles).
We appreciated this suggestion as very relevant, but we have already performed such studies before with the very same chemokine preps and have already published that data [Yoshida et al., 2014].
-) In the discussion, the authors state that the glycosylation of CCL2 in mammalian cells is important to stabilize the protein upon citrullination (something that is obviously not possible in bacteria). However, to state this point, authors should demonstrate that the glycosylation state of CCL2 is indeed important for its stability upon citrullination. For example, mutant CCL2 with mutated Asp 14 (which should not be glycosylated, to be verified and shown) could be tested against unmutated CCL2.
We highly appreciated this very logical suggestion by Reviewer 1. We performed a such site-directed mutagenesis experiment as shown in Fig. 4 with subsequent discussion of the data within the Revised manuscript.
Minor Points:
-) Spelling error at line 33 (important in for diagnostic).
We appreciate spelling corrections - changed for: “important in diagnostics”
-) Spelling error at line 211 (equivalent to completely processed to completely processed MCP-1/CCL2).
We appreciate the correction - changed for: “equivalent to completely processed MCP-1/CCL2”

Reviewer 2 Report
This manuscript "Mammalian glycosylation patterns protect citrullinated chemokine MCP1/CCL2 from partial degradation” addresses an important topic in the current field of inflammation and its diseases, particularly in rheumatoid arthritis. Therefore, these issues are of interest to the reader’s International Journal of Molecular Sciences. Although the results are of interest, the experimental approach and interpretation are generally problematic.
Figure 2 for citrullination of MCP1/CCL2 protein by using Mass-spectrometry and Figure 3 for self-made citrullinated protein by using ELISA are OK. However, figure 1 data did not support the statement that In vitro citrullination of bacterially produced chemokines leads to their partial degradation. It is also misconducted design experimentally. The issue is below.
à In general, human PAD2 protein size is about 70kDa. Figure 1a is wrong. I think that the transfected genes probably are human MCP2.
à PAD is known to convert arginine into citrulline in the target protein. When MCP/CCL2 protein was exposed to PAD, the citrullination of MCP1/CCL2 protein is naturally considered. Although total MCP1 is reduced in PAD2, it is not enough to explain and describe their degradation. Loading control is also lacking. If so, please show the data that ubiquitinated MCP1 protein following co-transfection and immunoprecipitation with MCP1.
Author Response
Dear Editor,
Dear Reviewers
We have greatly appreciated the very useful comments on our manuscript and performed additional experiments pointed out by the Reviewers. We believe, that we properly addressed all the Reviewers concerns and clearly explained all the unclear points. We hope, that the current version of our manuscript is greatly improved and now can be considered for publication in IJMS.
Specific concerns raised by Reviewers are pasted below with our replies point-by-point:
Reviewer 2
Comments and Suggestions for Authors
This manuscript "Mammalian glycosylation patterns protect citrullinated chemokine MCP1/CCL2 from partial degradation” addresses an important topic in the current field of inflammation and its diseases, particularly in rheumatoid arthritis. Therefore, these issues are of interest to the reader’s International Journal of Molecular Sciences. Although the results are of interest, the experimental approach and interpretation are generally problematic.
Figure 2 for citrullination of MCP1/CCL2 protein by using Mass-spectrometry and Figure 3 for self-made citrullinated protein by using ELISA are OK. However, figure 1 data did not support the statement that In vitro citrullination of bacterially produced chemokines leads to their partial degradation. It is also misconducted design experimentally. The issue is below.
à In general, human PAD2 protein size is about 70kDa. Figure 1a is wrong. I think that the transfected genes probably are human MCP2.
We absolutely agree with Reviewer 2’s statement about the size of human PAD2. We apologize for the confusion, and we changed a design of Figure 1 to make it clearer. We also added some additional text to Fig.1. legend: “Colloidal Coommasie stained MCP-1 bands are shown while recombinant humanPAD2/4 or rabbit PAD2 (Sigma aldrich) were added to reactions into the quantities below the Colloidal Coommasie sensitivity limits and cannot be visualized.” We believe, that now it is quite clear, that a part of a gel of 70kDa is not shown in the Figure and a label “PAD2/3/4” is used exclusively to show an exogeneously added enzyme used in the citrullination reaction.
A PAD is known to convert arginine into citrulline in the target protein. When MCP/CCL2 protein was exposed to PAD, the citrullination of MCP1/CCL2 protein is naturally considered. Although total MCP1 is reduced in PAD2, it is not enough to explain and describe their degradation. Loading control is also lacking.
We politely disagree with a Reviewer 2. Indeed, PAD enzymes convert a positively charged Arginine residue to a neutral (while still hydrophilic) residue, Citrulline. Such a conversion partially removes a positive charge of the protein and on the gel, such loss of a positively charged residue is indicated with a very characteristic gel shift of the citrullinated protein band due to its reduced motility in a standard SDS-PAGE system by Laemmli (please, see below an image of Fig. 7 in a paper by Lazarus RC et al, 2015.
… If so, please show the data that ubiquitinated MCP1 protein following co-transfection and immunoprecipitation with MCP1.
Shown in a Fig.1, we used a purified bacterially produced human recombinant MCP-1 – both self-made and a highly purified commercial one (R&D Systems). Similarly, the highly purified rabbit PAD2 (Sigma) or bacterially produced human recombinant PAD2/4 were used for in vitro citrullination reactions. Therefore, neither of mentioned reactions contained any traces of mammalian cellular extracts. On the other hand, bacteria possess a proteasome system, but proteasome-containing bacteria utilize a system termed pupylation that is functionally analogous to, but chemically distinct from, ubiquitylation [Jastrab & Darwin, 2015]. Therefore, we did not find this suggestion relevant to our study.
Figure 7. Specificity of anti-protein citrulline mAb 6B3 detection for Western blotting. Displayed on the left are two Coomassie-stained profiles showing the protein composition of native fibrinogen (Fib) and citrullinated fibrinogen (C-Fib). The Western blot (right) shows three immunoblots of which citrullinated fibrinogen was probed with active mAb 6B3 (Active/C-Fib; left lane); citrullinated fibrinogen was probed with immunoneutralized mAb 6B3 (Neutralized/C-Fib; middle lane); and native fibrinogen was probed with active mAb 6B3 (Active/Fib; right lane). - [ Lazarus RC et al, 2015.]
Cited Literature:
Lazarus Rachel C , John E Buonora 2 , Michael N Flora 3 , James G Freedy 3 , Gay R Holstein 4 , Giorgio P Martinelli 4 , David M Jacobowitz 5 , Gregory P Mueller . Protein Citrullination: A Proposed Mechanism for Pathology in Traumatic Brain Injury. // Front Neurol.. 2015 Sep 22;6:204. doi: 10.3389/fneur.2015.00204.
Jastrab Jordan B, Darwin K Heran. Bacterial Proteasomes // Annu Rev Microbiol., 2015;69:109-27. doi: 10.1146/annurev-micro-091014-104201.
